Susceptibility of lymnaeid snails to Fasciola hepatica and Fasciola gigantica (Digenea: Fasciolidae): a systematic review and meta-analysis

Ngcamphalala Philile Ignecious 1 212510872@stu.ukzn.ac.za
Nyagura Ignore 1
Malatji Mokgadi Pulane 1
http://orcid.org/0000-0001-6843-1497 Mukaratirwa Samson 1 2
1 School of Life Sciences, University of KwaZulu-Natal, Durban , Durban, KwaZulu-Natal , South Africa
2 One Health Centre for Zoonoses and Tropical Veterinary Medicine, Ross University School of Veterinary Medicine , Besseterre , Saint Kitts and Nevis
Avenant-Oldewage Annemariè
Electronic publication date: 2025 Mar 14
Publication date: 2025
Volume: 13
Electronic Location ID: e18976
Received 2024 Nov 12; Accepted 2025 Jan 21
Copyright: © 2025 Ngcamphalala et al.
Copyright year: 2025
Copyright holder: Ngcamphalala et al.
License: This is an open access article distributed under the terms of the Creative Commons Attribution License, which permits unrestricted use, distribution, reproduction and adaptation in any medium and for any purpose provided that it is properly attributed. For attribution, the original author(s), title, publication source (PeerJ) and either DOI or URL of the article must be cited.
License URL: https://creativecommons.org/licenses/by/4.0/

Keywords: Fasciola hepatica, Fasciola gigantica, Intermediate hosts, Lymnaeids, Experimental infections, Natural infections, Prevalence

Funding: European Union’s Horizon 2020 research and in-418 novation program 101000365 This work was supported by European Union’s Horizon 2020 research and in-418 novation program, grant number No. 101000365. The funders had no role in study design, data collection and analysis, decision to publish, or preparation of the manuscript.

==============================
Background

Fasciolosis is a food-borne disease that causes major economic losses, globally. This zoonotic disease is caused by Fasciola hepatica and Fasciola gigantica species which employ freshwater snails from the family Lymnaeidae as their intermediate hosts. Thus, a key aspect of understanding the epidemiology of the disease lies in understanding the transmission ecology of the parasite. Therefore, this systematic review and meta-analysis were conducted to assess the experimental susceptibility and prevalence of natural infections of F. hepatica and F. gigantica in lymnaeid snails.

Methods

Relevant peer-reviewed articles published in the past 20 years (2004–2023) were searched and appraised. Prevalence and infection rate estimates were based on 41 studies that met the inclusion criteria.

Results

Five thousand five hundred and seventy-five (5,575) lymnaeid snails were subjected to experimental infections and 44,002 were screened for natural infections. The overall pooled infection rate was higher in experimental infections 50% (95% CI [42–58%]) compared to natural infections of field-collected snails 6% (95% CI [0–22%]). The highest pooled infection rate was recorded in South America at 64% (95% CI [48–78%]) for experimental infections while the lowest was recorded for natural infections at 2% (95% CI [0–6%]) in Europe and 2% (95% CI [0–17%]) in Asia. In experimental studies, F. gigantica recorded the highest pooled prevalence at 73% (95% CI [61–84%] compared to F. hepatica which recorded 47% (95% CI [38–56%]). For natural infections, however, F. hepatica had the highest prevalence (12% (95% CI [0–30%]) while the lowest was noted for naturally infected F. gigantica at 2% (95% CI [0–18%]). Based on the snail species, the highest pooled prevalence was recorded for Pseudosuccinea columella infected with F. hepatica and F. gigantica at 47% (95% CI [33–61%]) while the lowest was recorded for F. hepatica naturally infected Galba truncatula at 4% (95% CI [0–10%]). Natural Fasciola spp. infections in intermediate snail hosts decreased in prevalence while experimental infections have increased in prevalence over the past 20 years.

Conclusions

While there seems to be a strong intermediate host specificity between the two Fasciola spp., experimental infection results showed that G. truncatula and R. natalensis are susceptible to F. hepatica and F. gigantica, respectively.

Introduction

Fasciolosis is a zoonotic food-borne disease of livestock and wild ruminants, and humans caused by the digenean liver flukes Fasciola hepatica Linnaeus, 1758 and Fasciola gigantica Cobbold, 1855 (Kithuka et al., 2002; Chen et al., 2013; Vázquez et al., 2019a; Ahmad et al., 2022). This parasitic disease causes economic losses in the livestock industry (Gutierrez et al., 2001) resulting from reduced productivity, liver condemnation, mortality, and expenditures for anthelmintics (Kithuka et al., 2002; Kleiman et al., 2007; Mucheka et al., 2015). These global economic losses have been estimated to exceed two billion dollars annually (Spithill, Smooker & Copeman, 1999; Medeiros et al., 2014; Lalor et al., 2021).

Fasciolosis has been recorded in more than 70 countries, across all inhabited continents (Lalor et al., 2021). Fasciola hepatica has been reported in temperate regions of Europe, Asia, Africa, Australia, and Americas (Mas-Coma, 2004; Lalor et al., 2021; Ahmad et al., 2022). Conversely, F. gigantica is mostly distributed in tropical and subtropical regions of Africa, Asia (Mas-Coma, 2004; Shoriki et al., 2014; Mucheka et al., 2015; Lalor et al., 2021; Ahmad et al., 2022), and the Middle East (Lalor et al., 2021). Furthermore, the overlapping geographical distribution of both species has been reported in many parts of Africa (Aleixo et al., 2015; Ahmad et al., 2022; Nukeri et al., 2022) and Central Asia where hybrid forms of the parasite have been recorded (Shoriki et al., 2014; Aleixo et al., 2015).

According to McKown & Ridley (1995), the presence of a compatible snail intermediate host (IH) in fasciolosis endemic areas is crucial to complete the life cycle of the parasite. The transmission of Fasciola spp. in a specific geographical region mostly depends on the presence of vector snails from the family Lymnaeidae Rafinesque, 1815 (Alba et al., 2019). As Vinarski (2013) reported, this family consists of two subfamilies, Radicinae and Lymnaeinae, with about 26 genera collectively. Species from the genera Galba (Cruz-Mendoza et al., 2004; Gutierrez et al., 2011; Alemu, 2019), Lymnaea (Kim et al., 2014; Alemu, 2019), Pseudosuccinea (Cucher et al., 2006; Alemu, 2019; Ngcamphalala, Malatji & Mukaratirwa, 2022), Forassia (Cruz-Mendoza et al., 2004; Alemu, 2019), Radix (Bargues et al., 2011; Imani-Baran et al., 2012; Huang et al., 2019), Austropeplea (Dung et al., 2013; Kim et al., 2014), and Omphiscola (Correa et al., 2017; Rondelaud, Vignoles & Dreyfuss, 2022) act as the IHs for Fasciola species. Although approximately 1,200 lymnaeid snail species have been described globally (Vázquez et al., 2019b), only 30 species are known as IHs of Fasciola spp. (Alba et al., 2019; Vázquez et al., 2019a). These species are distributed worldwide (Prepelitchi et al., 2011; Vinarski, 2013) and have been reported to extend from tropical to temperate regions with some occurring at extremely cold latitudes (Vázquez et al., 2019b).

Like most trematodes, Fasciola spp. show a marked snail host specificity (Bargues & Mas-Coma, 2005; Bargues et al., 2011). Fasciola gigantica is mainly transmitted by snail species from the genus Radix, particularly those belonging to Hubendick’s (1951) superspecies of Radix auricularia (Linnaeus, 1758), which includes R. natalensis (Krauss, 1848) in Africa, R. rubiginosa (Michelin, 1831) in Asia (Brown, 1994) and R. auricularia in Europe (Mas-Coma, Bargues & Valero, 2005). Recently, Pseudosuccinea columella has been shown to transmit this species in Africa (Grabner et al., 2014, Carolus et al., 2019; Malatji & Mukaratirwa, 2020). Fasciola hepatica is, however, transmitted by snails from various genera, but Galba species are the main IHs of this species globally (Bargues & Mas-Coma, 2005). According to Vázquez et al. (2019a), research on the compatibility of snail and parasite populations may serve to understand better and predict the transmission of fasciolosis globally. Additionally, a key aspect of understanding the epidemiology of a disease lies in understanding the transmission ecology of the parasite (Vázquez et al., 2015). Furthermore, understanding the susceptibility of the IHs to Fasciola spp. in a given locality may assist with the development of strategic control programs. Therefore, this study reviewed and analysed the results of peer-reviewed research reporting on the global experimental susceptibility/infectivity and natural infections of F. hepatica and F. gigantica in lymnaeid snail species in the past 20 years (2004–2023).

Methods

Search strategy

A systematic literature search was conducted on PubMed, Web of Science, and Google Scholar databases, and a combination of the following search terms and Boolean operators (OR, AND) were used: Fasciola hepatica OR Fasciola gigantica AND intermediate hosts OR lymnaeids OR Lymnaeidae OR Pseudosuccinea OR Galba OR Fossaria OR Lymnaea OR Omphiscola OR Austropeplea OR Radix AND natural infection OR experimental infection OR prevalence OR infectivity. Peer-reviewed articles published in the last 20 years (2004–2023) were retrieved and appraised. Additional articles were identified by cross-referencing selected articles’ biographies (snowballing). EndNote reference manager version X8 (Clarivate Analytics, Philadelphia, PA, USA) was used to retrieve and manage full-text articles.

Inclusion and exclusion criteria

The following inclusion criteria were used to select articles for the systematic review and meta-analysis: (i) clearly stated the number of lymnaeid snails screened and/or infected with F. hepatica and F. gigantica, (ii) identified and reported the Fasciola species up to species level, (iii) identified the intermediate host snails up to species level, (iv) reported prevalence based on natural infections or infection rate based on experimental infections, and (v) indicated the detection method used.

Studies were excluded if they reported only on Fasciola spp. infection in definitive hosts, identification and distribution of the intermediate hosts without Fasciola infections, and articles written in other languages besides English.

Data extraction

Based on the study design, PIN and IN screened the titles and abstracts of articles, and relevant articles were retrieved. Full texts of retrieved articles were thoroughly reviewed and those that did not meet the inclusion criteria were excluded. Microsoft (MS) Excel spreadsheet was used to capture data from the text, tables, and figures for meta-analysis. Data extracted from relevant articles included the author’s names, year of publication, continent, country where the study was conducted, Fasciola species, snail host, sample size, number of infected host snails, prevalence, and detection method. In cases of differences regarding the inclusion and exclusion criteria, the researchers discussed and reached a favorable conclusion, however, the final decision was made by Dr. MP Malatji.

Quality assessment

The overall quality of the articles for meta-analysis was assessed using the Grading of Recommendations Assessment, Development, and Evaluation (GRADE) approach (Guyatt et al., 2008; Doi & Thalib, 2008). The quality of all studies included was assessed by scoring one point for each inclusion criterion that was fulfilled and a 0 was given for each unfulfilled criterion. As a result, each selected study was assigned a score ranging from 0 to 5. Studies with an index score of five were deemed high quality, four as moderate, and those that scored ≤3 were considered low quality and excluded from the analysis (Doi & Thalib, 2008). The total quality scores for all included studies ranged from moderate to good quality (Tables S1, S2).

Data analysis

The double arcsine approach was used to transform the prevalence data to avoid overestimating the weight of individual studies (Barendregt et al., 2013). This approach utilizes the arcsine transformation twice to the prevalence values to account for the heterogeneity caused by studies with extreme proportions or smaller sample sizes (Barendregt et al., 2013). MetaXL add-in for Microsoft Excel (www.epigear.com) was employed to compute a quality effects model to account for the heterogeneity (Barendregt et al., 2013). The level of heterogeneity between estimates was evaluated using inverse variance statistic (I2 index), and the differences were accounted for using Cochrane’s Q test (Barendregt, 2016). The I2 index score was interpreted as low heterogeneity if it was <25%, moderate at 50%, and high heterogeneity at >75% (Higgins et al., 2003). The estimated prevalence and the 95% confidence interval (CI) of Fasciola species infections in lymnaeid snails were demonstrated on forest plots. Subgroup analysis was done to assess heterogeneity and factors that could influence the observed pooled prevalence estimates (PPE); thus, the data was grouped according to the region/continent on which studies were conducted, snail species involved, Fasciola species, method of detection, and period covered by the studies and publication bias was evaluated using funnel plots. To identify the sources of heterogeneity, meta-regression was performed with continents, diagnostic tests, and Fasciola species and study period fixed as independent factors. The meta-regression was treated as a linear model on the logit-transformed prevalence data. Using Egger’s test, the linear regression analysis was conducted to evaluate publication bias.

Results

Search results

Of the 861 articles obtained after going through the search databases and snowballing, 774 articles were excluded because they were either duplicates or deemed ineligible based on the title and/or abstract contents (Fig. 1). Full-texts of 46 studies were retrieved and assessed for eligibility based on the predetermined inclusion criteria and five articles were deemed ineligible. The remaining 41 studies met the inclusion criteria and quality assessment. These were distributed across Europe (13/41; 31.7%), Africa (10/41; 24.4%), South America (9/41; 21.9%), Asia (8/41; 19.5%), and North America (1/41; 2.43%). Of these studies, 46.3% (19/41) reported experimental infection while 53.7% (22/41) reported natural infections.

Figure 1 PRISMA diagram.

Overall experimental susceptibility of lymnaeid snails to Fasciola spp.

Experimental infections of Fasciola species in lymnaeid snails were conducted in Europe (France and Sweden), and in Africa (Egypt). In Asia, experiments were conducted in Iran while in South America experiments were conducted in Argentina, Colombia, and Cuba (Table S1). A total of 5,575 freshwater lymnaeid snails were collected to be subjected to experimental infections with Fasciola spp. between 2004 and 2023. Of these freshwater snails, 2,815 (50.45%) were infected with either F. hepatica or F. gigantica. The lymnaeid snail species involved in the experiments were Galba (G.) truncatula, G. cubensis, G. cousini, G. viatrix var ventricose, G. neotropica, Pseudosuccinea (P.) columella, Radix (R.) auricularia, R. natalensis, Lymnaea (L.) fuscus, L. palustris, and Omphiscola (O.) glabra. Experimental infection rates ranged from 7.89% in L. fuscus infected with F. hepatica to 80.37% in G. cubensis infected with F. gigantica (Table 1). Of the 11 snail species mentioned above, nine were successfully infected with F. hepatica and four with F. gigantica. Only P. columella and G. truncatula had records of successful experimental infections with both Fasciola species (Table 1). The overall pooled experimental infection rate of lymnaeid snails with Fasciola spp. was 50% (95% CI [42–58%] (Fig. S1). High heterogeneity in the results was revealed by the quality effects model (Q = 1,425, p < 0.001), with I2 = 96% (Fig. S1). The meta-regression model demonstrated a statistically significant overall p-value of 0.05 for experimental investigations, suggesting that the predictors collectively account for the variation in prevalence (Table 2). The R-squared change was 0.119, indicating that 11.9% of the variance in the prevalence data could be explained by the predictors. However, the prevalence level was not significantly impacted by individual factors (Table 2).

Table 1 Frequency of lymnaeid snails experimentally infected with F. gigantica and F. hepatica in the past 20 years.

Snail species	No. of studies	No. infected	No. examined	Diagnostic tool (%)	Species of infection	Overall prevalence (%)	
				Shedding	Dissection	Fasciola hepatica	Fasciola gigantica		
Galba (G.) truncatula	10	1,196	2,155	72.03	53.47	1,155	41	55.49	
G. cubensis	1	217	270	–	80.37	–	217	80.37	
G. cousini	1	34	100	–	34.00	34	–	34.00	
G. neotropica	1	50	159	–	31.45	50	–	31.45	
Radix (R.) auricularia	1	115	151	76.16	–	–	115	76.16	
R. natalensis	2	83	308	–	26.95	83	–	26.95	
Pseudosuccinea (P.) columella	8	945	1,963	40.71	52.02	848	97	48.14	
Lymnaea (L.) fuscus	1	9	114	–	7.89	9	–	7.89	
L. palustris	1	40	119		33.61	40	–	33.61	
L. viatrix var. ventricosa	1	73	133	–	54.89	73	–	54.89	
Omphiscola (O.) glabra	1	53	103	–	51.46	53	–	51.46	
Total	–	2,815	5,575	52.74	49.81	2,386	429	50.45	

Table 2 Meta-regression of overall and subgroups for individual variables on prevalence of Fasciola infections in snail intermediate host in the past 20 years.

Model	Unstandardized coefficients	Standardized coefficients	t	Sig.	R2	95.0% Confidence interval for B	
B	Std. Error	Beta	Lower bound	Upper bound	
Natural infection	Continents	0.124	0.364	0.069	0.341	0.736		−0.622	0.870	
Fasciola sp.	1.616	0.964	0.362	1.676	0.105		−0.362	3.595	
Diagnostic test	−0.411	0.617	−0.116	−0.666	0.511		−1.677	0.855	
period	−1.883	0.639	−0.458	−2.947	0.007		−3.195	−0.572	
	Combined effect					0.001	0.474			
Experimental infection	Continents	0.167	0.136	0.234	1.227	0.226		−0.107	0.441	
	Fasciola sp.	−0.533	0.476	−0.182	−1.120	0.269		−1.493	0.427	
	Diagnostic test	−0.033	0.344	−0.015	−0.097	0.923		−0.727	0.661	
	period	0.672	0.332	−0.391	2.021	0.050		0.002	1.343	
	Combined effect					0.05	0.194			

Experimental infectivity of lymnaeid snails by Fasciola spp. by continent

The highest pooled infection rate for Fasciola spp. (Figs. 2A–2C) was 64% recorded in South America (95% CI [48–78%], Fig. 2C), followed by Africa at 48% (95% CI [35–61%], Fig. 2B) and Europe at 42% (95% CI [28–56%], Fig. 2A). Asia did not qualify for the meta-analysis. Heterogeneity was recorded at I2 = 96% for all three reported continents.

Figure 2 Forest plots of experimental infection rates of Fasciola hepatica and Fasciola gigantica in lymnaeid snails from (A) Europe, (B) Africa, and (C) South America (Alba et al., 2018; Dar et al., 2010, 2013, 2014b, 2015b; Dreyfuss, Vignoles & Rondelaud, 2012, 2016; Novobilský et al., 2013; Pointier et al., 2007; Rondelaud et al., 2004, 2015; Salazar, Estrada & Velásquez, 2006; Sanabria et al., 2012, 2013; Vázquez et al., 2014; Vignoles, Dreyfuss & Rondelaud, 2015).

Different letters next to references denote different data generated from the same publication.

Experimental infection rate of lymnaeid snails by Fasciola spp. per snail species

Only three of the 11 recorded lymnaeid snail species qualified for meta-analysis, viz G. truncatula, R. natalensis, and P. columella. The estimated pooled infection rate of lymnaeid snails infected with F. hepatica and F. gigantica is illustrated in Figs. 3A–3C. Galba truncatula infected with F. hepatica (96.57%, 1,155/1,196) and F. gigantica (3.43%, 41/1,196) (Table 1) showed a pooled infection rate of 37% (95% CI [17–59%], Fig. 3A), R. natalensis with F. hepatica with a prevalence of 21% (95% CI [03–48%], Fig. 3C) and P. columella with both F. hepatica (43.20%, 848/1,963) and F. gigantica (4.64%, 91/1,963) (Table 1) with a pooled infection rate of 47% (95% CI [33–61%], Fig. 3B). Galba truncatula and P. columella showed a heterogeneity of I2 = 97% while R. natalensis demonstrated a heterogeneity of I2 = 95%.

Figure 3 Forest plots of experimental infection rates of Fasciola species based on the intermediate hosts: (A) Galba truncatula, (B) Pseudosuccinea columella, and (C) Radix natalensis (Alba et al., 2018; Dar et al., 2010, 2013, 2014b, 2015b; Dreyfuss, Vignoles & Rondelaud, 2012, 2016; Novobilský et al., 2013; Pointier et al., 2007; Rondelaud et al., 2004, 2015; Sanabria et al., 2012, 2013; Salazar, Estrada & Velásquez, 2006; Vázquez et al., 2014; Vignoles, Dreyfuss & Rondelaud 2015).

Different letters next to references denote different data generated from the same publication.

Experimental infection rate of lymnaeid snails by parasite species

The pooled infection rate of Fasciola spp. to lymnaeid snails was high with F. gigantica at 73% (95% CI [61–84%] and low in F. hepatica at 47% (95% CI [38–55%]. However, heterogeneity was higher for F. hepatica (Q = 1,234.09, p < 0.001; I2 = 96%, Fig. 4A) compared to F. gigantica (Q = 31.73, p < 0.001; I2 = 87%, Fig. 4B).

Figure 4 Forest plots showing experimental infection rates in lymnaeid snails based on Fasciola species: (A) Fasciola hepatica and (B) Fasciola gigantica (Alba et al., 2018; Ashrafi & Mas-Coma, 2014; Dar et al., 2010, 2013, 2014b, 2015b; Dreyfuss, Vignoles & Rondelaud, 2012, 2016; Novobilský et al., 2013; Pointier et al., 2007; Rondelaud et al., 2015, 2004; Salazar, Estrada & Velásquez, 2006; Sanabria et al., 2012, 2013; Vázquez et al., 2014; Vignoles, Dreyfuss & Rondelaud 2015).

Different letters next to references denote different data generated from the same publication.

Experimental infectivity of lymnaeid snails by Fasciola spp. per method of detection

The pooled experimental infection rate of Fasciola snails in lymnaeid snails was higher using cercariae shedding 52% (95% CI [32–72%], Fig. 5A, Table 1) compared to 49% (95% CI [40–59%], Fig. 5B, Table 1) using snail dissection. Heterogeneity was documented as I2 = 97% and I2 = 96% for shedding and dissection, respectively.

Figure 5 Forest plots of experimental infection rates of Fasciola hepatica and Fasciola gigantica based on detection technique (A) shedding and (B) dissection (Alba et al., 2018; Ashrafi & Mas-Coma, 2014; Dreyfuss, Vignoles & Rondelaud, 2016, 2012; Dar et al., 2010, 2013, 2014b, 2015b; Novobilský et al., 2013; Pointier et al., 2007; Rondelaud et al., 2004, 2015; Salazar, Estrada & Velásquez, 2006; Sanabria et al., 2012, 2013; Vázquez et al., 2014; Vignoles, Dreyfuss & Rondelaud 2015).

Different letters next to references denote different data generated from the same publication.

Experimental infection rate of lymnaeid snails by Fasciola spp. based on years

The estimated pooled infection rate of Fasciola spp. experimentally infected snails for 20 years is shown in Fig. S2. The pooled infection rate in the decade 2004–2013 was 41% (95% CI [29–53%], Fig. S2A), which was lower than the 57% (95% CI [47–68%], Fig. S2B) pooled infection rate documented between 2014–2023. Heterogeneity was I2 = 96% for both periods (Figs. S2A–S2B).

Publication bias of studies reporting on experimental infections of Fasciola spp. in lymnaeid snails

Figure S3 shows the funnel plot which is asymmetrical in shape and depicts publication bias which may result from either a small sample size or publication bias within articles.

Natural infections of lymnaeid snails by Fasciola spp.

Field prevalence data for lymnaeid snails infected by Fasciola spp. was recorded in South America (Colombia, Cuba, Ecuador, and Argentina), North America (Mexico), Africa (Egypt and South Africa), Europe (Ireland, Sweden, Spain, Poland, and France) and Asian (Iran, India, South Korea, China, and Vietnam) (Table S2). Of the 44,002 field-collected lymnaeid snails, 5,656 were positive for Fasciola infections with an overall prevalence of 12.85%. Prevalence of lymnaeid snails naturally infected with F. hepatica and F. gigantica ranged from 0–76.9% in L. fuscus and G. bulimoides, respectively (Table 3). The 17 infected snail species were G. truncatula, G. cousini, G. viatrix, G. humilis, G. schirazensis, G. bulimoides, P. columella, R. cucunorica, R. gedrosiana, R. peregra, R. auricularia, R. acuminata, L. palustris, L. ollula, L. fuscus, O. glabra, and Austropeplea (A.) viridis (Table 3). Only P. columella and G. truncatula, however, qualified for meta-analysis. The estimated overall pooled prevalence for natural infections in lymnaeid snails was recorded at 6% (95% CI [0–22%]) (Fig. S4). A significantly high heterogeneity was recorded Q = 15,220.37 (p < 0.001), with I2 = 100% (Fig. S4). The R-squared change was 0.474, and the p-value was significant at 0.001 (Table 2). This suggests that the predictors have a greater impact and that the model accounts for 47.4% of the variation in prevalence. Regarding each of the specific factors, the p-values for the continents, diagnostic tests, and Fasciola species were all greater than 0.05 for natural snail infection by Fasciola, indicating no significant effects. Nonetheless, the p-value for the time frame was 0.007, suggesting that time had a significant effect on the prevalence of Fasciola infection (Table 2).

Table 3 Frequency of lymnaeid snails naturally infected with F. gigantica and F. hepatica in the past 20 years.

Snail species	No. of studies	No. examined	No. infected	Diagnostic tool (%)		Species of infection	Overall prevalence
(%)	
				Dissection	Molecular	Shedding	RFLP	Fasciola hepatica	Fasciola gigantica		
Galba (G.) truncatula	7	8,617	408	3.25	12.87	–	–	408	–	4.73	
G. bulimoides	1	670	515	76.87	–	–	–	515	–	76.87	
G schirazensis	2	1,517	96	–	6.33	–	–	96	–	6.33	
G. humilis	1	3,372	2,537	75.24	–	–	–	2,537	–	75.24	
G. viatrix	1	68	22	–	61.76	2.94	–	22	–	32.35	
G. cousini	1	521	68	13.05	–	–	–	68	–	13.05	
Radix (R.) acuminata	2	2,477	250					–	250	10.09	
R. auricularia	1	496	12	–	–	–	2.45	–	12	2.45	
R. peregra	1	167	62	–	37.13	–	–	62	–	37.13	
R. gedrosiana	1	2,543	298	–	11.72	–		–	298	11.72	
R. cucunorica	1	409	179	–	43.77	–		179	–	43.77	
Pseudosuccinea (P.) columella	4	976	278	–	32.51	17.25	–	168	110	28.48	
Lymnaea (L.) fuscus	1	130	0	–	0	–	–	–	–	0	
R. ollula	1	15	5	–	33.33	–	–	5	–	33.33	
L. palustris	1	668	1	–	0.15	–	–	1	–	0.15	
Omphiscola glabra	2	5,980	102	1.26	21.54	–	–	102	–	1.71	
Austropeplea (A.) viridis	2	15,376	125	–	0.81	–	–	1	124	0.81	
Total	–	44,002	5,656	20.72	5.81	15.69	2.45	4,576	1,080	12.85	

Prevalence of natural infections of lymnaeid snails by Fasciola spp. per continent

The estimated pooled prevalence estimates for lymnaeid snails naturally infected by Fasciola spp. globally is illustrated in Figs. 6A–6C. The prevalence recorded per continent was 2% in Asia (95% CI [0–17], Fig. 6A), 2% in Europe (95% CI [0–6%], Fig. 6B), and 11% in South America (95% CI [0–29%], Fig. 6C). Africa and North America data did not qualify for meta-analysis. Heterogeneity results were I2 = 98%, I2 = 99%, and I2 = 100% for South America, Europe, and Asia, respectively (Figs. 6A–6C).

Figure 6 Forest plots of the infection rates of natural infections of Fasciola species in lymnaeid snails based on continents; (A) Asia, (B) Europe, and (C) South America (Caron et al., 2017; Correa et al., 2017; Cucher et al., 2006; Dung et al., 2013; Geli-Erazo et al., 2020; Gutierrez et al., 2011; Huang et al., 2019; Iglesias-Piñeiro et al., 2016; Imani-Baran et al., 2012; Kim et al., 2014; Kozak & Wędrychowicz, 2010; Martínez-Ibeas et al., 2013; Novobilský et al., 2013; Pereira, Uribe & Pointier, 2020; Rajanna et al., 2018; Relf et al., 2009; Rondelaud, Vignoles & Dreyfuss, 2022; Sunita et al., 2021; Yakhchali, Malekzadeh-Viayeh & Imani-Baran, 2014).

Different letters next to references denote different data generated from the same publication.

Prevalence of natural infections of lymnaeid snails by Fasciola spp. per snail species

Pseudosuccinea columella and G. truncatula were the only snail species that qualified for meta-analysis. The average prevalence of natural infections of P. columella infected with F. hepatica was 17.21% (168/976) and F. gigantica was 11.27% (110/976) (Table 3), with an overall pooled prevalence of 26% (95% CI [0–72%], Fig. 7A). Galba truncatula infected by F. hepatica recorded a pooled prevalence of 4% (95% CI [0–10%], Fig. 7B). Heterogeneity results were documented as I2 = 99% for both lymnaeid species.

Figure 7 Forest plots of rates of natural infections of Fasciola species based on intermediate snail host (A) Pseudosuccinea columella and (B) Galba truncatula (Malatji & Mukaratirwa, 2020; Grabner et al., 2014; Gutierrez et al., 2011; Cucher et al., 2006; Martínez-Ibeas et al., 2013; Kozak & Wędrychowicz, 2010; Iglesias-Piñeiro et al., 2016; Correa et al., 2017; Rondelaud, Vignoles & Dreyfuss, 2022; Arafa et al., 2018; Pereira, Uribe & Pointier, 2020).

Different letters next to references denote different data generated from the same publication.

Prevalence of natural infections of lymnaeid snails by Fasciola spp. per parasite species

The estimated pooled infection rates of the individual Fasciola spp. naturally infecting lymnaeid snails were reported in Figs. 8A–8B. Fasciola gigantica showed a prevalence of 2% (95% CI [0–18%], Fig. 8A) and Fig. 8B shows a prevalence of 12% (95% CI [0–30%] for F. hepatica natural infections. The heterogeneity was significantly high for both F. gigantica (Q = 1,873.85, p < 0.001, I2 = 100%) and F. hepatica (Q = 11,616.37, p < 0.001, I2 = 100%) natural infections (Figs. 8A–8B).

Figure 8 Forest plots showing the rates of infection of Fasciola spp. in naturally infected lymnaeid snails based on species (A) Fasciola gigantica, and (B) Fasciola hepatica (Malatji & Mukaratirwa, 2020; Grabner et al., 2014; Yakhchali, Malekzadeh-Viayeh & Imani-Baran, 2014; Sunita et al., 2021; Rajanna et al., 2018; Imani-Baran et al., 2012; Dung et al., 2013; Relf et al., 2009; Novobilský et al., 2013; Martínez-Ibeas et al., 2013; Kozak & Wędrychowicz, 2010; Iglesias-Piñeiro et al., 2016; Correa et al., 2017; Rondelaud, Vignoles & Dreyfuss, 2022; Arafa et al., 2018; Kim et al., 2014; Huang et al., 2019; Pereira, Uribe & Pointier, 2020; Gutierrez et al., 2011; Geli-Erazo et al., 2020; Caron et al., 2017; Cruz-Mendoza et al., 2004; Cucher et al., 2006).

Different letters next to references denote different data generated from the same publication.

Prevalence of natural infections of lymnaeid snails by Fasciola spp. per method of detection

Detection of natural Fasciola spp. infections in lymnaeid snails was based on molecular (PCR), restriction fragment length polymorphism (RFLP), dissection, and cercariae shedding techniques (Table 3, Table S2). Prevalence based on the diagnostic tool utilized raged from 20.72% by dissection to 2.45% by RFLP (Table 3). Molecular and dissection techniques were the only techniques that qualified for meta-analysis (Figs. 9A–9B). Low pooled prevalence was recorded with molecular technique (4%, 95% CI [0–17%] (Fig. 9A) compared to dissection technique at 12% (95% CI [0–40%] (Fig. 9B). Recorded heterogeneity was I2 = 100% for dissection and I2 = 99% for molecular technique (Fig. 9B).

Figure 9 Forest plots showing the rates of natural infections of Fasciola species in lymnaeid snails recorded using (A) molecular techniques and (B) dissection (Arafa et al., 2018; Sunita et al., 2021; Relf et al., 2009; Novobilský et al., 2013; Kozak & Wędrychowicz, 2010; Correa et al., 2017; Malatji & Mukaratirwa, 2020; Grabner et al., 2014; Kim et al., 2014; Imani-Baran et al., 2012; Huang et al., 2019; Dung et al., 2013; Gutierrez et al., 2011; Caron et al., 2017; Cucher et al., 2006; Rajanna et al., 2018; Geli-Erazo et al., 2020; Pereira, Uribe & Pointier, 2020; Cruz-Mendoza et al., 2004; Martínez-Ibeas et al., 2013; Iglesias-Piñeiro et al., 2016; Rondelaud, Vignoles & Dreyfuss, 2022).

Different letters next to references denote different data generated from the same publication.

Prevalence of natural infections of lymnaeid snails by Fasciola spp. by years

Pooled prevalence for natural Fasciola spp. infections in their IHs hosts was 9% (95% CI [0–70%], Fig. S5A) in the decade 2004–2013, which was higher than 3% (95% CI [0–9%] (Fig. S5B) in 2014–2023 (Figs. S5A–S5B). Heterogeneity was I2 = 100% for 2004–2013 (Fig. S5B) and I2 = 99% for 2014–2023 (Fig. S5B).

Publication bias of studies reporting on the natural infections of Fasciola spp. in lymnaeid snail spp

Funnel plots showed an asymmetric funnel shape (scattered points) (Fig. S6) indicating the presence of publication bias which may be due to content in the articles or small sample size.

Discussion

The results showed that the overall pooled prevalence of natural infections of Fasciola species in lymnaeids was significantly lower than the infection rate recorded based on experimental infections. This was to be expected as conditions for experimental infections are made optimum and controlled in laboratory settings compared to natural environments where various uncontrolled variables and stressors may hinder successful infection of the intermediate host. Infections under laboratory settings have been shown to be a valuable approach in the investigation of compatibility differences in IHs since they allow for the control and management of variables that can influence the infection outcome (Sorensen & Minchella, 2001; Vázquez et al., 2019a). These variables include snail shell size (Dar et al., 2014a; Dreyfuss, Vignoles & Rondelaud, 2016), the infective parasite dose (Sorensen & Minchella, 2001; Pointier et al., 2007), optimum conditions (optimum temperature, constant light/dark periods, abundant food, pollution free water, dissolved essential minerals) (de Kock, van Eeden & Pretorius, 1986; Dar et al., 2014a, 2015a; Dreyfuss, Vignoles & Rondelaud, 2016), deliberate exposure to miracidia (Dreyfuss, Vignoles & Rondelaud, 2016; Ashrafi & Mas-Coma, 2014) amongst others. This control of variables leads to a higher infection rate as the probability of one snail getting infected strictly depends on its suitability as IH and its surrounding ecology (Vázquez et al., 2015). Prepelitchi et al. (2011) also noted that snail populations are subjected to rigorous ecological constraints due to large environmental temporal fluctuations. Additionally, parasites may die before finding an appropriate IH in a natural environment, especially those with a narrow tolerance to specific physicochemical factors (Vázquez et al., 2015).

Overall, while the model for natural infection demonstrates stronger explanatory power (R-square = 0.474), the mixed results point to the complexity of factors influencing Fasciola infection prevalence in snails, which requires further investigation with more refined models or additional data. Nyagura et al. (2024) claimed that when several factors are combined, the variability becomes much more prevalent, suggesting that understanding how these factors interact is essential to comprehending the complexity of epidemiological results. Instead of a single determinant, the results of this study corroborate the idea that the epidemiology of snail-borne parasites is typically driven by a confluence of factors that interact significantly (Hajipour et al., 2021).

The wide global distribution of lymnaeid snails is of great concern as the geographic distribution of these Fasciola spp. depends on the availability and ecological needs of their respective intermediate host species (Malatji, Lamb & Mukaratirwa, 2019). As expected, this review showed a range of lymnaeid snails that were implicated in the transmission of Fasciola species in the field and in experimental setting in five of the six inhabited continents. These results are consistent with previous reports as this snail family has been reported in all continents except Antarctica (Vázquez et al., 2019b). For both experimental infections and natural infections, South America recorded the highest pooled prevalence, and the lowest prevalence was recorded in Europe and Asia. While all three continents recorded multiple snail species involved in the natural transmission of F. hepatica, G. truncatula and O. glabra contributed more to the pooled prevalence. Mas-Coma, Bargues & Valero (2005), implicated the geographical expansion of G. truncatula and P. columella in the dispersal of F. hepatica from Europe to other continents. However, though infections in G. truncatula were also noted in South America, other Galba species (G. schirazensis) proved to contribute more to the pooled prevalence. The latter results may be a misinterpretation however, as G. schirazensis had been frequently confused with G. truncatula. A reason for this may be that G. schirazensis and G. truncatula can be considered as cryptic species as they are very similar in anatomical variation and shell morphology (Correa et al., 2011; Vázquez et al., 2019a). Furthermore, South America comparatively recorded the highest infections (76.9%) of Fasciola spp. (Cruz-Mendoza et al., 2004) compared to Europe (38.3%) (Martínez-Ibeas et al., 2013).

Fasciola gigantica recorded a high pooled prevalence in experimental infections, however, in the natural environment, the pooled prevalence was higher with F. hepatica. Furthermore, the later species showed a wider geographical expansion, recorded in five continents while F. gigantica showed restriction to two continents. Contributing to this may be the wide range of snail species involved in the natural transmission of F. hepatica, which explains its geographical expansion compared to F. gigantica. Furthermore, Mas-Coma, Bargues & Valero (2005) linked the smaller geographical distribution of F. gigantica to its IHs having a weaker diffusion capacity.

Our analysis showed that while various snail species have been subjected to experimental infections and assessed for natural infection of Fasciola spp., most studies were conducted on G. truncatula, R. natalensis, and P. columella. Our results further showed that the invasive snail, P. columella, which recorded a high pooled prevalence was infected by both F. hepatica and F. gigantica for both experimental and natural infections, despite G. truncatula and R. natalensis being the main IHs of F. hepatica (Gasnier et al., 2000; Cruz-Mendoza et al., 2004; Bargues & Mas-Coma, 2005; Pointier et al., 2009; Kim et al., 2014; Beesley et al., 2018; Alemu, 2019) and F. gigantica (Bargues & Mas-Coma, 2005; Mas-Coma, Bargues & Valero, 2005; Rajanna et al., 2018; Alemu, 2019; Nyagura, Malatji & Mukaratirwa, 2022), respectively. The ability of P. columella to transmit both F. hepatica and F. gigantica in a natural environment has been documented in many countries (Grabner et al., 2014; Carolus et al., 2019; Alba et al., 2019; Malatji & Mukaratirwa, 2020; Ngcamphalala, Malatji & Mukaratirwa, 2022). Furthermore, according to Mas-Coma, Bargues & Valero (2005), P. columella has been linked to the secondary transmission of F. hepatica. Additionally, A. viridis was shown to transmit both F. hepatica (Kim et al., 2014) and F. gigantica (Dung et al., 2013) in nature even though the prevalence of both infections was significantly low. However, it has been previously noted that even low prevalence in naturally infected intermediate snails matters (Vázquez et al., 2015) as a single miracidium infection can produce approximately 4,000 metacercariae leading to substantial environmental contamination (Andrews, 1999; Nguyen et al., 2012).

Experimental studies recorded a high pooled prevalence based on cercariae shedding, which is the most used and affordable detection method to assess trematode infection in snails. However, this method tends to underestimate the true prevalence of infection as it mainly detects patent infections, and infections that are still at the prepatent stage are regarded as negatives (Curtis & Hubbard, 1990; Born-Torrijos et al., 2014; Tigga et al., 2014). Whilst microscopic dissection recorded the lowest pooled prevalence in experimental infections, this method however had the highest prevalence in natural infections with molecular detection recording the lowest. Contributing to the lowest prevalence of molecular detection (polymerase chain reaction (PCR) might have been due to the low number of studies using this technique to detect Fasciola spp. infections in IH snails. Furthermore, most studies might have opted not to use molecular techniques due to the costs involved equipment, consumables, and sequencing (Caron et al., 2010; Tigga et al., 2014), and the lack of skilled personnel to carry out the PCR (Tigga et al., 2014). Several authors emphasized investing in PCR as a supplementary/confirmation method as it has been shown to detect Fasciola spp. infections (DNA) in IH snails after they had been deemed negative by shedding and/or dissection (Caron et al., 2017; Rajanna et al., 2018; Geli-Erazo et al., 2020; Malatji & Mukaratirwa, 2020).

Experimental infections showed a high pooled infection rate of Fasciola spp. infections in the most recent decade (2014–2023) which might be attributed to the increased number of experimental studies conducted to better understand the host–parasite interaction and transmission of these important zoonotic parasites. However, prevalence data for natural infections showed a decline in prevalence for the past 20 years, with the decade (2014–2023) recording the lowest pooled prevalence. The natural infection model’s significant results for the period (p = 0.007) indicate that temporal factors, such as seasonal variation or changes in environmental conditions, may be crucial in influencing prevalence, despite the overall heterogeneity seen across the various factors (continents, diagnostic tests, and Fasciola species). This decline in natural infections may be attributed to climate change as the primary determinant of transmission efficiency is the relationship between rainfall and temperature (Fox et al., 2011) both of which have either positive or negative effects on the distribution of the intermediate snail hosts and survival of free-living stages of the Fasciola spp. parasite (Madsen & Stauffer, 2022). Another reason for the decline in natural infections prevalence may be the effectiveness of control strategies targeting infections in definitive hosts and IHs such as the use of anthelmintics and controlling the snail IH using molluscicides (Madsen & Stauffer, 2022).

There could be a number of reasons for the continents, diagnostic tests, and Fasciola species’ lack of significance, such as the fact that these variables may not vary sufficiently between studies or that other unmeasured confounders may obscure their impact. The observed high level of heterogeneity may potentially indicate that the results are being influenced by other factors that were not taken into account in the model, such as methodological or regional variances.

The limitation of this review is that only articles written in English were included to ensure that there was no misrepresentation of methodologies and results in cases where there could be incorrect translations from other languages to English. Additionally, publication bias was detected for both field and experimental studies. Despite using a standardized analysis process, it was difficult to achieve consistent meta-analysis due to the differences in study design, detection, and quantification methods in the different studies. Furthermore, several studies failed to provide complete information on the prevalence of Fasciola species amongst freshwater snails, and those that had all the information were not evenly distributed across continents. Hence, meta-analysis could not be conducted for some IH species and some continents. As a result, the prevalence data presented in this review does not fully represent the prevalence of Fasciola spp. infections amongst freshwater snail spp. globally.

Conclusions

The review highlighted crucial information on the prevalence of Fasciola spp. infection in their intermediate snail hosts across the globe. Natural infection results showed a strong intermediate host specificity between the two Fasciola spp., where G. truncatula and R. natalensis are susceptible to F. hepatica and F. gigantica respectively, whilst P. columella is able to transmit both species. This information is important in determining and estimating the species-specific distribution and transmission, which can be used as baseline data for interrupting the life cycle of fasciolosis in a given area.

We therefore, recommend continuous surveillance and monitoring of the dispersal of the lymnaeid snail species involved in the transmission of specific Fasciola spp. Additionally, to employ the use of molecular detection methods to supplement classic parasitological methods, to confirm the detection of infection and identity of species. Moreover, focus on developing and using other protocols such as the loop-mediated isothermal amplification (LAMP) and other PCR-based protocols that can detect and identify species without the extra sequencing costs is required.

Supplemental Information

Supplemental Information 1 PRISMA checklist.

Supplemental Information 2 Supplementary figures.

Supplemental Information 3 Supplementary tables.

Supplemental Information 4 Audience intended.

Additional Information and Declarations

Competing Interests

The authors declare that they have no competing interests.

Author Contributions

Philile Ignecious Ngcamphalala performed the experiments, analyzed the data, prepared figures and/or tables, authored or reviewed drafts of the article, and approved the final draft.

Ignore Nyagura performed the experiments, analyzed the data, prepared figures and/or tables, and approved the final draft.

Mokgadi Pulane Malatji conceived and designed the experiments, analyzed the data, prepared figures and/or tables, authored or reviewed drafts of the article, resolved disagreements on search strategy, and approved the final draft.

Samson Mukaratirwa conceived and designed the experiments, authored or reviewed drafts of the article, and approved the final draft.

Data Availability

The following information was supplied regarding data availability:

This is a systematic review/meta-analysis.

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
