# Peer review of "Susceptibility of lymnaeid snails to Fasciola hepatica and Fasciola gigantica (Digenea: Fasciolidae): a systematic review and meta-analysis"

_PeerJ, doi:10.7717/peerj.18976_

## Round 0.1 · original submission · Major Revisions

Reviewer 2 has a number of questions about the validity and repeatability of the data. The authors need to address these concern and provide additional information ( if that is applicable) in the review manuscript and also in a rebuttal giving point by point feedback on how they addressed the comments.

Reviewer 1 highlighted a huge inconsistency between the manuscript and the reference list.

Both reviewers provide editorial input for improvement of the writing.

Reviewer 1 ·

Basic reporting

The manuscript is written in good, passable English, although I have proposed rephrasing of some sentences in a few instances (see attached pdf). Relevant literature is cited sufficiently to cover the study content. However, 26 extra references are listed that do not appear in text citations. I'm not sure if this can be related to errors in the reference manager software (as is sometimes the case), or the authors simply failed to thoroughly check the list. This is so glaring.

The review article is well structured and the results are relevant, bringing out an important picture on the global prevalence of fascioliasis from experimental studies and natural infections. The results adequately satisfy the research hypothesis.

Experimental design

The review study is well designed using standard, well-justified methods. and the study is within the scope and aims of the journal. The meta-analysis is technically sound and all data collection was standardised optimally, despite difficulties with the variability of the literature assessed.

Validity of the findings

The data was meaningfully presented and analysed. Tables 1 and 2 can benefit by adding columns showing the prevalence for each parasite species, as well as the overall prevalence for each method.

Additional comments

I have added a few comments in the attached annotated pdf.

Annotated reviews are not available for download in order to protect the identity of reviewers who chose to remain anonymous.

Reviewer 2 ·

Basic reporting

no comment

Experimental design

no comment

Validity of the findings

no comment

Additional comments

Main Comments
1. The study mentions that different studies employed various detection and quantification methods for Fasciola spp. infections. This variability can lead to inconsistencies in results, making it difficult to compare findings across studies. The lack of a standardized method raises questions about the reliability of the data collected from different sources. How would the author explain it?
2. The research acknowledges that despite using a standardized analysis process, achieving consistent meta-analysis was challenging due to differences in study design, detection, and quantification methods across various studies. This variability can lead to questions about the reliability of the pooled results. How would the author explain it?
3. Why is there such a significant difference between the pooled infection rates of experimental infections (50%) and natural infections (6%)? What specific controlled conditions were maintained during experimental infections that differed from natural environments? The research highlights those experimental infections showed significantly higher pooled infection rates than natural infections, suggesting that laboratory conditions may not accurately reflect real-world scenarios. Understanding these controlled conditions could provide insights into the factors influencing infection rates.
4. How were environmental factors accounted for in experimental and natural infection studies? The research indicates that uncontrolled variables may affect infection rates, yet it does not clarify how these were managed.
5. What statistical methods were used to analyze the data, and how were heterogeneity issues addressed? The high heterogeneity observed in the results, particularly for F. hepatica, raises questions about the appropriateness of the statistical methods used. Clarifying the statistical approach could help in understanding the reliability of the pooled infection rates.
6. What specific details are provided to ensure that the methodologies described are replicable by other researchers? While the abstract mentions sufficient detail for replication, further elaboration on the methods used would enhance transparency.
7. 5,575 lymnaeid snails were subjected to experimental infections; how were these snails selected, and do they represent the broader population? The representativeness of the sample could significantly influence the generalizability of the findings.
8. How were the detection methods for Fasciola spp. infections standardized across the studies included in the meta-analysis? Variability in detection methods could lead to inconsistencies in reported infection rates, raising concerns about the reliability of the data.
9. What steps were taken to address the high heterogeneity observed in the results, particularly for F. hepatica? The reported heterogeneity (I² = 96%) suggests that the results may not be consistent across studies, which could undermine the validity of the conclusions drawn from the meta-analysis.

Minor Comments/Corrections
*Line 18 “was” should be “were”
*Line 31 “when compared” should be “compared.”
*Line 37 “a decrease” should be “decreased.”
*Line 38, “shown an increase in prevalence,” should be “have increased”
*Line 56 “Asia” should be “and Asia.”
*Line 63 “allow for the completion of” should be “complete.”
*Line 65-66 “As reported by” should be “As Vinarski (2013) reported”
*Line 92 “analysed” should be “analyzed”
*Line 97 “A systematic search of literature” should be “A systematic literature search.”
*Line 103-104 “through and cross-referencing the bibliographies of selected articles” should be “and cross-referencing selected articles' bibliographies.”
*Line 108 “the titles and abstracts of articles were screened by PIN and IN” should be “PIN and IN screened the titles and abstracts of articles, and relevant articles were retrieved.”
*Line 122 “the country” should be “country”
*Line 133 “as high” should be “high”
*Line 142-144 “Using inverse variance statistic (I2 index) the level of heterogeneity between estimates was evaluated and the differences were accounted for using Cochrane’s Q test (Barendregt and Doi, 2016).” should be “The level of heterogeneity between estimates was evaluated using inverse variance statistic (I2 index), and the differences were accounted for using the Cochrane’s Q test (Barendregt and Doi, 2016).”
*Line 165 “while in South America in Argentina, Colombia, and Cuba” should be “while in South America, experiments were performed in Argentina, Colombia, and Cuba”
*Line 167 “, and” should be “to”
*Line 172 “above-mentioned snail species” should be “snails mentioned above species”
*Line 177 “was” should be “were”
*Line 243 “was” should be “were”
*Line 263 “which indicates” should be “indicates the publication bias due to the article content”
*Line 169 “as compared” should be “compared.”
*Line 272 “dependent on” should be “depends on”
*Line 274-276 “Additionally, the possibility of parasites dying before finding an appropriate IH in a natural environment is higher, especially in those with a narrow tolerance to specific physicochemical factors” should be “Additionally, parasites may die before finding an appropriate IH in a natural environment, especially those with a narrow tolerance to specific physicochemical factors”
*Line 279 “results from this review,” should be “this review showed”

Annotated reviews are not available for download in order to protect the identity of reviewers who chose to remain anonymous.

---

## Round 0.2 · accepted · Accept

The authors have addressed the reviewers comments and the manuscript is ready for publication. I confirmed that the concerns of Reviewer 1 have been attended to.

Reviewer 2 ·

Basic reporting

no comment

Experimental design

no comment

Validity of the findings

no comment

Additional comments

To the Editor and authors
I have reviewed all nine main points of explanation provided by the authors. The author has addressed and incorporated additional information into the manuscript, covering all aspects comprehensively and adjusting the minor points. Thank you for the explanations and revisions based on my suggestions, and I hope these recommendations and modifications will further enhance the completeness of this manuscript.